# SHE$^2$: Stochastic Hamiltonian Exploration and Exploitation for Derivative-Free Optimization

## Abstract

Derivative-free optimization (DFO) using trust region methods is frequently used for machine learning applications, such as (hyper-)parameter optimization without the derivatives of objective functions known. Inspired by the recent work in continuous-time minimizers, our work models the common trust region methods with the exploration-exploitation using a dynamical system coupling a pair of dynamical processes. While the first exploration process searches the minimum of the blackbox function through minimizing a time-evolving surrogation function, another exploitation process updates the surrogation function time-to-time using the points traversed by the exploration process. The efficiency of derivative-free optimization thus depends on how the two processes couple. In this paper, we propose a novel dynamical system, namely SHE$^2$—$\underline{\text{S}}$tochastic $\underline{\text{H}}$amiltonian $\underline{\text{E}}$xploration and $\underline{\text{E}}$xploitation, that surrogates the subregions of blackbox function using a time-evolving quadratic function, then explores and tracks the minimum of the quadratic functions using a fast-converging Hamiltonian system. The SHE$^2$ algorithm is later provided as a discrete-time numerical approximation to the system. To further accelerate optimization, we present P-SHE$^2$ that parallelizes multiple SHE$^2$ threads for concurrent exploration and exploitation. Experiment results based on a wide range of machine learning applications show that P-SHE$^2$ outperform a boarder range of derivative-free optimization algorithms with faster convergence speed under the same settings.

## 1 Introduction

Derivative-free optimization (DFO) techniques (Powell, 1964), such as Bayesian optimization algorithms (Snoek et al., 2012; Martinez-Cantin, 2014), non-differentiable coordinate descent (Colson & Toint, 2002), natural gradient method (Grosse & Salakhudinov, 2015; Hasenclever et al., 2017), and natural evolution strategies (Schaul et al., 2011), have been widely used for black-box function optimization. DFO techniques have been viewed as one of promising solutions, when the first-order/higher-order derivatives of the objective functions are not available. For example, to train large-scale machine learning models, parameter tuning is sometimes required. The problem to find the best parameters from the high-dimensional parameter space is frequently formalized as a black-box optimization problem, as the function that maps the specific parameter settings to the performance of models is not known (Golovin et al., 2017; Fazel et al., 2018; Xu et al., 2018b; Liu et al., 2018). The evaluation of the black-box function is often computationally expensive, and there thus needs DFO algorithms to converge fast with global/local minimum guarantee.

**Backgrounds.** To ensure the performance of DFO algorithms, a series of pioneering work has been done (Conn et al., 2009; Powell, 2012; Jamieson et al., 2012; Augustin & Marzouk, 2017; Golovin et al., 2017). Especially, Powell *et al.* (Powell, 1964; 1994) proposed *Trust-Region* methods that intends to "surrogate" the DFO solutions through exploring the minimum in the trust regions of the blackbox objective functions, where the trust regions are tightly approximated using model functions (e.g., quadratic functions or Gaussian process) via interpolation. Such two processes for *exploration* and *exploitation* are usually alternatively iterated, so as to pursue the global/local minimum (Augustin & Marzouk, 2017). With *exploration* and *exploitation* (Dezza et al., 2017), a wide range of algorithms have been proposed using trust region for DFO surrogation (Powell, 2002; Queipo et al.,

2005; Snoek et al., 2012; Swersky et al., 2013; Schulman et al., 2015; Wu et al., 2017; Regier et al., 2017; Abdolmaleki et al., 2017; Osokin et al., 2017; Lyu et al., 2018; Arenz et al., 2018).

**Technical Challenges.** Though trust region methods have been successfully used for derivative-free optimization for decades, the drawbacks of these methods are still significant:

- *The computational and storage complexity for (convex) surrogates is extremely high.* To approximate the trust regions of blackbox functions, *quadratic functions* (Powell, 2002; Regier et al., 2017) and *Gaussian process* (Snoek et al., 2012; Wu et al., 2017; Lyu et al., 2018) are frequently used as (convex) surrogates. However, fitting the quadratic functions and Gaussian process through interpolation is quite time-consuming with high sample complexity. For example, using quadratic functions as surrogates (i.e., approximation to the second-order Taylor's expansion) needs to estimate the gradient and inverse Hessian matrix (Powell, 2002; Regier et al., 2017), where a large number of samples are required to avoid ill-conditioned inverse Hessian approximation; while the surrogate function in GP is nonconvex, which is even more sophisticated to optimize.

- *The convergence of trust region methods cannot be guaranteed for high-dimensional non-convex DFO.* Compared to the derivative-based algorithms such as stochastic gradient descent and accelerated gradient methods (Bottou, 2010; Su et al., 2014), the convergence of DFO algorithms usually are not theoretically guaranteed. Jamieson et al. (Jamieson et al., 2012) provided the lower bound for algorithms based on boolean-based comparison of function evaluation. It shows that DFO algorithms can converge at $\Omega(1/\sqrt{T})$ rate in the best case ($T$ refers to the total number of iterations), without assumptions on convexity and smoothness, even when the evaluation of black-box function is noisy.

**Our Intuitions.** To tackle the technical challenges, we are motivated to study novel trust region methods with following properties

1. *Low-complexity Quadratic Surrogates with Limited Memory.* To lower the computational complexity, we propose to use quadratic functions with identity Hessian matrices as surrogates. Rather than incorporating all evaluated samples in quadratic form approximation, our algorithm only works with the most-recently evaluated sample points. In this way, the memory consumption required can be further reduced. However, the use of identity Hessian matrices for quadratic form loses the information about the distribution (e.g., Fisher information or covariance (Hansen et al., 2003)) of evaluated sample points.

2. *Fast Quadratic Exploration with Stochastic Hamiltonian Dynamical Systems.* Though it is difficult to improve the convergence rate of the DFO algorithms in general nonconvex settings with less oracle calls (i.e., times of function evaluation), one can make the exploration over the quadratic trust region even faster. Note that exploration requires to cover a trust region rather than running on the fastest path (e.g., the gradient flow (Hu & Li, 2017)) towards the minimum of trust region. In this case, there needs an exploration mechanism traversing the whole quadratic trust region in a fast manner and (asymptotically) approaching to the minimum. Figure 1 illustrates the examples of exploration processes over the quadratic region via its gradient flows (i.e., gradient descent) or using Hamiltonian dynamics with gradients (Neal et al., 2011) as well as their stochastic variants with explicit perturbation, all in the same length of time. It shows that the stochastic Hamiltonian dynamics (shown in Figure 1(d)) can well balance the needs of fast-approaching the minimum while sampling the quadratic region with its trajectories. Compared to the (stochastic) gradient flow, which leads to the convergence to the minimum in the fast manner, the stochastic Hamiltonian system are expected to well explore the quadratic trust region with the convergence kept. Inspired by theoretical convergence consequences of Hamiltonian dynamics with Quadratic form (Su et al., 2014; Neal et al., 2011), we propose to use stochastic Hamiltonian dynamical system for exploring the quadratic surrogates.

3. *Multiple Quadratic Trust Regions with Parallel Exploration-Exploitation.* Instead of using one quadratic cone as the surrogate, our method constructs the trust regions using multiple quadratic surrogates, where every surrogate is centered by one sample point. In this way, the information of multiple sample points can be still preserved. Further, to enjoy the speedup of parallel computation, the proposed method can be accelerated through exploring the minimum from multiple trust regions (using multiple Hamiltonian dynamical sys-

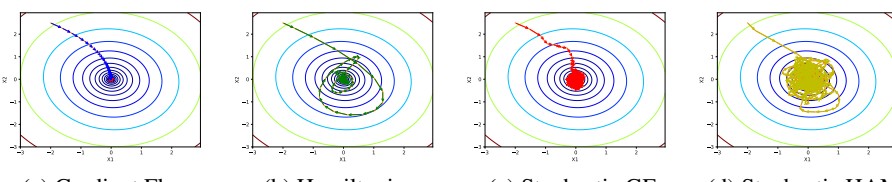

| (a) Gradient Flow | (b) Hamiltonian | (c) Stochastic GF | (d) Stochastic HAM |

Figure 1: Quadratic Surrogate Exploration via Gradient Flow, Hamiltonian Dynamics with Gradient, Stochastic Gradient Flow, and Stochastic Hamiltonian Dynamics with Gradient

> tems) concurrently while parallelizing the blackbox function evaluation to save the overall time. Certain synchronization among the Hamiltonian systems might be required to share knowledge among the parallel exploration-exploitation processes.

Our work is inspired by the recent progress in the continuous-time convex minimizers (Su et al., 2014; Hu & Li, 2017; Xu et al., 2018a) on convex functions, where the optimization algorithms are considered as the discrete-time numerical approximation to some (stochastic) ordinary differential equations (ODEs) or dynamics, such as Itô processes for SGD algorithms (Hu & Li, 2017) or Hamiltonian systems for Nesterov's accelerated SGD (Su et al., 2014). We intend to first study the new ODE and dynamical system as a continuous-time DFO minimizer that addresses above three research issues. With the new ODE, we aim at proposing the discrete-time approximation as the algorithms for black-box optimization.

**Our Contributions.** Specifically, we make following contributions. **(1)** To address the three technical challenges, a continuous-time minimizer for derivative-free optimization based on a Hamiltonian system coupling two processes for exploration and exploitation respectively. **(2)** Based on the proposed dynamical system, an algorithm, namely $SHE^2$–$\underline{S}$tochastic $\underline{H}$amiltonian $\underline{E}$xploration and $\underline{E}$xploitation, as a discrete-time version of the proposed dynamical system, as well as P-$SHE^2$ that parallelizes $SHE^2$ for acceleration. **(3)** With the proposed algorithms, a series of experiments to evaluate $SHE^2$ and P-$SHE^2$ using real-world applications. The two algorithms outperform a wide range of DFO algorithms with better convergence. To the best of our knowledge, this work is the first to use a Hamiltonian system with coupled process for DFO algorithm design and analysis.

## 2 RELATED WORK AND PRELIMINARIES

In this section, we first review the most relevant work of trust region methods for DFO problem, then present the preliminaries of this work.

### 2.1 TRUST REGION METHODS FOR DFO

The trust region algorithms can be categorized by the model functions used for surrogates. Generally, there are two types of algorithms adopted: *Gaussian Process (GP)* (Snoek et al., 2012; Swersky et al., 2013; Wu et al., 2017; Lyu et al., 2018) or *Quadratic functions* (Powell, 2002; Regier et al., 2017; Osokin et al., 2017) for surrogation. Blessed by the power of Bayesian nonparameteric statistics, Gaussian process can well fit the trust regions, with confidence bounds measured, using samples evaluated by the blackbox function. However, the GP-based surrogation cannot work in high dimension and cannot scale-up with large number of samples. To solved this problem, GP-based surrogation algorithms using the kernel gradients (Wu et al., 2017) and mini-batch (Lyu et al., 2018) have been recently studied.

On the other hand, the quadratic surrogation (Powell, 2002) indeed approximates the trust region through interpolating the second-order Taylor expansion of the blackbox objective. With incoming points evaluated, there frequently needs to numerically estimate and adapt the inverse Hessian matrix and gradient vector, which is extremely time-consuming and sample-inefficiency (with sample complexity $\mathcal{O}(d^2)$ for $d$-dimensional DFO (Powell, 2002)). Following such settings, (Regier et al., 2017) proposed to a second-order algorithm for blackbox variational inference based on quadratic surrogation, while (Osokin et al., 2017) leveraged a Gaussian Mixture Model (multiple quadratic surrogations) to fit the policy search space over blackbox probabilistic distribution for policy optimization. A novel convex model generalizing the quadratic surrogation has been recently proposed to characterize the loss for structured prediction (Arenz et al., 2018).

In addition, some evolutionary strategies, such as Covariance Matrix Adaptation Evolution Strategy (CMA-ES) (Hansen et al., 2003; Abdolmaleki et al., 2017), indeed behave as a sort of quadratic surrogate as well. Compared to the common quadratic surrogate, CMA-ES models the energy of blackbox function using a multivariate Gaussian distribution. For every iteration, CMA-ES draws a batch of multiple samples from the distribution, then statistically updates parameters of the distribution using the samples with blackbox evaluation. CMA-ES can be further accelerated with parallel blackbox function evaluation and has been used for hyperparameter optimization of deep learning (Loshchilov & Hutter, 2016).

## 2.2 HAMILTONIAN SYSTEMS FOR CONVEX MINIMIZATION

Here, we review the Nesterov's accelerated method for quadratic function minimization. We particularly are interested in the ODE of Nesterov's accelerated method and interpret behavior of the ODE as a Hamiltonian dynamical system.

**Corollary 1** (ODE of Nesterov's Accelerated Method). *According to Su et al. (2014), the discrete-time numerical format of the Nesterov's accelerated method Nesterov (2004) can be viewed as an ODE as follow.*

$$\ddot{Z}(t) + \frac{3}{t}\dot{Z}(t) + \nabla_Z f(Z(t)) = 0, \tag{1}$$

*where $f(X)$ is defined as the objective function for minimization and $\nabla_X f(Z(t))$ refers to the gradient of the function on the point $Z(t)$. Above ODE can converge with strongly theoretical consequences if the function $f(X)$ is convex with some smoothness assumptions Su et al. (2014).*

**Corollary 2** (Convergence of Eq 1 over Quadratic Loss). *Let's set $f(X) = \frac{1}{2}\|X - X^*\|_2^2$ According to the ODE analysis of Nestereov's accelerated method Su et al. (2014), the ODE listed in Eq 1 converges with increasing time $t$ at the following rate:*

$$\|Z(t) - X^*\|_2^2 \leq \frac{4\|Z(0) - X^*\|_2^2}{t^2} = \mathcal{O}\left(\frac{1}{t^2}\right), \tag{2}$$

*where $X(0)$ refers to the initial status of the ODE. The proof has been given in Su et al. (2014).*

## 3 STOCHASTIC HAMILTONIAN SYSTEM FOR BLACK-BOX OPTIMIZATION

In this section, we present the proposed Hamiltonian system for Black-Box minimization via exploration and exploitation. Then, we introduce the algorithms and analyze its approximation to the dynamical systems.

### 3.1 THE STOCHASTIC HAMILTONIAN EXPLORATION AND EXPLOITATION SYSTEM

Given a black-box objective function $f(X)$ and $X \in \mathbb{R}^d$, we propose to search the minimum of $f(X)$ in the $d$-dimensional vector space $\mathbb{R}^d$, using a novel Hamiltonian system, derived from the ODE of Nesterov's accelerated method and Eq 1, yet without the derivative of $f(X)$ needed.

**Definition 1** (Quadratic Loss Function). *Given two $d$-dimensional vectors $X$ and $Y$, we characterizes the Euclid distance between the two vectors using the function as follow.*

$$Q(X, Y) = \frac{1}{2}\|X - Y\|_2^2, \tag{3}$$

*where the partial derivative of $Q$ on $X$ should be $\frac{\partial}{\partial X}Q(X, Y) = X - Y$ indicating the fastest direction moving from $X$ to $Y$.*

**Definition 2** (Stochastic Hamiltonian Exploration and Exploitation). *As was shown in Eq. 4, a Hamiltonian system is designed with following two coupled processes:* exploration process $X(t)$ *and* exploitation process $Y(t)$, *where $t$ refers to the searching time. These two processes are coupled withn each other. Specifically, the exploration process $X(t)$ in Eq 4 uses a second order ODE to track the dynamic process $Y(t)$, while the exploiting process $Y(t)$ always memorizes the minimum point (i.e., $X(\tau)$) that have been reached by $X(t)$ from time 0 to $t$.*

$$\begin{cases} \textbf{(a):} & \ddot{X}(t) + \frac{3}{t}\dot{X}(t) + \frac{\partial}{\partial X}Q(X(t), Y(t)) + \zeta(t) = 0, \\ \textbf{(b):} & Y(t) = X(\tau) \text{ and } \tau = \underset{\tau \in (0,t]}{\operatorname{argmin}} f(X(\tau)), \end{cases} \tag{4}$$

*where (1) $\frac{\partial}{\partial X}Q(X(t), Y(t)) = X(t) - Y(t)$ indicates the fastest direction to track $Y(t)$ from $X(t)$; and (2) the perturbation term $\zeta(t)$ referring to an unbiased random noise with controllable bound*

---

**Algorithm 1** SHE$^2$: Stochastic Hamiltonian Exploration and Exploitation

---

1: $X_0$, $V_0 \sim$ randomly from $\mathbb{R}^d$ and $Y_0 \leftarrow X_0$ /* initialization by randomization*/;
2: **for** $t = 1, 2, \ldots, T$ **do**
3:     /* SHE$^2$ $X(t)$ Process Update*/
4:     $\beta_t \leftarrow -(\alpha_t + 3/t)$ and $\gamma_t \leftarrow 3/t$;
5:     $\zeta_t \overset{iid}{\sim} \mathcal{N}(0, I)$ and $\zeta_t \leftarrow \varepsilon \cdot \zeta_t / |\zeta_t|_2$;
6:     $X_t \leftarrow X_{t-1} + \alpha_t \cdot V_{t-1}$;
7:     $V_t \leftarrow V_{t-1} + \alpha_t \cdot Y_{t-1} + \beta_t \cdot X_{t-1} + \gamma_t \cdot X_t + \alpha_t \cdot \zeta_t$;
8:     /* SHE$^2$ $Y(t)$ Process Update*/
9:     **if** $f(X_t) \leq f(Y_{t-1})$ **then**
10:         $Y_t \leftarrow X_t$;
11:     **else**
12:         $Y_t \leftarrow Y_{t-1}$;
13:     **end if**
14: **end for**
15: **return** $Y_T$;

---

*$\|\zeta(t)\|_2 \leq \epsilon$ would help the system escape from an unstable stationary point in even shorter time. In the above dynamical system, we treat $Y(t)$ as the minimizer of the black-box function $f(X)$.*

Indeed, SHE$^2$ approximates the black-box function $f(X)$ using a simple yet effective quadratic function, then leverages the ODE listed in Eq 1 to approximate the minimum with the quadratic function. With the new trajectories traversed by Eq 1, the quadratic function would be updated. Through repeating such *surrogation-approximation-updating* procedures, the ODE continuously tracks the time-dependent evolution of quadratic loss functions and finally stops at a stationary point when the quadratic loss functions is no longer updated (even with new trajectories traversed).

**Remark 1.** *We can use the analytical results Su et al. (2014) to interpret the dynamical system (in Eq 4) as an adaptive perturbated dynamical system that intends to minimize the Euclid distance between $X(t)$ and $Y(t)$ at each time $t$. The memory complexity of this continuous-time minimizer is $\mathcal{O}(1)$, where a Markov process $Y(t)$ is to used to memorize the status quo of local minimum during exploration and exploitation.*

**Theorem 1** (Convergence of SHE$^2$ Dynamics)**.** *Let's denote $x^*$ as a possible local minimum point of the landscape function $f(x)$. We have as $t \to \infty$, with high probability, that $X(t) \to x^*$, where $X(t)$ is the solution to* (4).

Please refer to the Lemma 1 and Lemma 2 in the Appendix for the proof of above theorems. We will discuss the rate of convergence, when introducing SHE$^2$ algorithm as a discrete-time approximation to SHE$^2$.

## 3.2 SHE$^2$: ALGORITHM DESIGN AND ANALYSIS

Given a black-box function $f(x)$ and a sequence of non-negative step-size $\alpha_t$ (t=0, 1, 2, \ldots, T), which is small enough, as well as the scale of perturbation $\varepsilon$, we propose to implement SHE$^2$ as Algorithm 1. The output of algorithm $Y_T$ refers to the value of $Y_t$ in the last iteration (i.e., the $t^{th}$ iteration). The whole algorithm only uses the evaluation of function $f(x)$ for comparisons, without computing its derivatives. In each iteration, only the variable $Y_t$ is dedicated to memorize the local minimum in the sequence of $X_1$, $X_2$, \ldots, $X_t$. Thus the memory complexity of SHE$^2$ is $\mathcal{O}(1)$.

In terms of convergence, Jamieson et al Jamieson et al. (2012) provided an universal lower bound on the convergence rate of DFO based on the "boolean-valued" comparison of (noisy) function evaluation. SHE$^2$ should enjoy the same convergence rate $\Omega(1/\sqrt{T})$ without addressing any further assumptions.

## 3.3 APPROXIMATION ANALYSIS FOR SHE$^2$

Here, we would demonstrate that the proposed algorithm behaves as a discrete-time approximation to the dynamical systems of $X(t)$ and $Y(t)$ addressed in Eq 4, while as $\alpha_t \to 0$ the sequence of $X_t$ and $Y_t$ (for $1 \leq t \leq T$) would converge to the behavior of continuous-time minimizer — coupled processes $X(t)$ and $Y(t)$.

Given an appropriate constant step-size $\alpha_t \to 0$ for $t = 1, 2..., T$, we can rewrite the the sequences $X_t$ described in lines 4–7 of Algorithm 1 as the following Stochastic Differential Equation (SDE) of $X^\omega(t)$ with the random noise $\omega(t)$:

$$
\begin{cases}
\dfrac{X_t - X_{t-1}}{\alpha_t} = V_{t-1} & \stackrel{\alpha_t \to 0}{\Longrightarrow} \quad \dot{X}(t) = V(t), \\[2ex]
\dfrac{V_t - V_{t-1}}{\alpha_t} = -\dfrac{3(X_t - X_{t-1})}{t \cdot \alpha_t} & \stackrel{\alpha_t \to 0}{\Longrightarrow} \quad \dot{V}(t) = -\dfrac{3}{t}X(t) - (X(t) - Y(t)) + \zeta(t), \\[1ex]
\qquad\qquad + (Y_{t-1} - X_{t-1}) + \zeta_t &
\end{cases}
\tag{5}
$$

where $\zeta(t)$ refers to the continuous-time dynamics of sequence $\zeta_1, \zeta_2, \ldots, \zeta_T$ and $|\zeta(t)|_2 = \varepsilon$ for every time $t$. Through combining above two ODEs and **Lemma** 1, we can obtain the SDE of $X(t)$ based on the perturbation $\zeta(t)$ as:

$$
\ddot{X}(t) + \frac{3}{t}\dot{X}(t) + \frac{\partial}{\partial X}Q(X(t), Y(t)) + \zeta(t) = 0.
\tag{6}
$$

The sequence $Y_t$ (t=0, 1, 2, $\cdots T$) always exploits the minimum point that has been already found by $X_t$ at time $t$. Thus, we can consider $Y_t$ is the discrete-time of $Y(t)$ that exploits the minimum traversed by $X(t)$. In this way, we can consider the coupled sequences of $X_t$ and $Y_t$ (for $1 \leq t \leq T$) as the discrete-time form of the proposed dynamical system with $X(t)$ and $Y(t)$.

# 4 P-SHE$^2$: PARALLEL STOCHASTIC HAMILTONIAN EXPLORATION AND EXPLOITATION

To enjoy the speedup of parallel computation, we propose a new Hamiltonian dynamical system with a set of ODEs that leverage multiple pairs of coupled processes for exploration and exploitation in parallel. Then, we present the algorithm design as a discrete-time approximation to the ODEs.

## 4.1 THE P-SHE$^2$ DYNAMICAL SYSTEM

Given a black-box objective function $f(X)$ and $X \in \mathbb{R}^d$, we propose to search the minimum of $f(X)$ in the $d$-dimensional vector space $\mathbb{R}^d$, using following systems.

**Definition 3** (Parallel Stochastic Hamiltonian Exploration and Exploitation). *As was shown in Eq. 7, a Hamiltonian system is designed with (1) $N$ pairs of coupled exploration-exploitation processes: $X^i(t)$ and $Y^i(t)$ for $1 \leq i \leq N$ that explores and exploits the minimum in-parallel from $N$ (random/unique) starting points, and (2) an overall exploitation process $Y(t)$ memorizing the local minimum traversed by the all $N$ pairs of coupled processes. Specifically, for each pair of coupled processes, a new surrogation model $Q_\delta(X^i(t), Y^i(t), Y(t))$ has been proposed to measure the joint distance from $X^i(t)$ to $Y^i(t)$ and $Y(t)$ respectively, where $\delta > 0$ refers to a trade-off factor weighted-averaging the two distances.*

$$
\begin{cases}
\textbf{(a):} \quad \ddot{X}^i(t) + \dfrac{3}{t}\dot{X}^i(t) + \dfrac{\partial}{\partial X}Q_\delta\left(X^i(t), Y^i(t), Y(t)\right) + \zeta^i(t) = 0 \\[1.5ex]
\qquad\quad Q_\delta(X, Y, Z) = \delta Q(X, Y) + (1 - \delta)Q(X, Z) \\[1.5ex]
\textbf{(b):} \quad Y^i(t) = X^i(\tau) \text{ and } \tau = \underset{\tau \in (0,t]}{\operatorname{argmin}} f(X^i(\tau)), \qquad \forall 1 \leq i \leq N \\[2ex]
\textbf{(c):} \quad Y(t) = X^j(\tau) \text{ and } j = \underset{1 \leq j \leq N}{\operatorname{argmin}} f(Y^i(\tau))
\end{cases}
\tag{7}
$$

*where $\frac{\partial}{\partial X}Q_\delta(X^i(t), Y^i(t), Y(t)) = X^i(t) - \delta \cdot Y^i(t) - (1 - \delta) \cdot Y(t)$ indicates the fastest direction to track $Y^i(t)$ and $Y(t)$, jointly, from $X^i(t)$. In the above dynamical system, we treat $Y(t)$ as the minimizer of the black-box function $f(X)$.*

**Remark 2.** *We understand the dynamical system listed in Eq 7 as a perturbated dynamical system with multiple state variables, where all variables are coupled to search the minimum of $f(X)$ through $X^i(t)$ (for $1 \leq i \leq N$). The memory complexity of this continuous-time minimizer is $\mathcal{O}(N)$, where every Markov process $Y^i(t)$ is to used to memorize the status quo of local minimum traversed by the corresponding processes.*

## 4.2 P-SHE$^2$: SCALING-UP SHE$^2$ THROUGH PARALLELIZED FUNCTION EVALUATION

The evaluation of black-box function $f(x)$ is frequently time consuming. **Algorithm** 2 presents an algorithm namely P-SHE$^2$that uses $N$ SHE$^2$ threads to evaluate $f(x)$ in parallel. Specifically, in

each iteration, P-SHE$^2$ leverages the minimum among the $N$ local minimums searched by the $N$ parallel SHE$^2$ threads to accelerate the overall search. In P-SHE$^2$, we use $N$ threads of SHE$^2$ to search the minimum, where each thread uses three $X_t^j$, $Y_t^j$ and $V_t^j$ trajectories for exploration and exploitation. The memory complexity of P-SHE$^2$ is $\mathcal{O}(N)$.

---

**Algorithm 2** P-SHE$^2$: Stochastic Hamiltonian Exploration and Exploitation

1:  /* Starting P-SHE$^2$ by Initializing $N$ instances of SHE$^2$ with Synchronization*/
2:  **for** $j = 1, 2, 3, \ldots, N$ **do**
3:      $X_0^j, V_0^j \sim$ randomly from $\mathbb{R}^d$ and $Y_0^j \leftarrow X_0^j$;
4:  **end for**
5:      $\underset{\forall X \in \{Y_0^1, Y_0^2, \ldots, Y_0^N\}}{\text{argmin}} \quad f(X) \rightarrow Y_0$;
6:  **for** $t = 1, 2 \ldots, T$ **do**
7:      **for** $j = 1, 2, 3, \ldots, N$ **in Parallel do**
8:          /* $X(t)$ update for the $j^{th}$ SHE$^2$ thread*/
9:          $\beta_t \leftarrow -(\alpha_t + 3/t)$ and $\gamma_t \leftarrow 3/t$;
10:         $\zeta_t^j \overset{iid}{\sim} \mathcal{N}(0, I)$ and $\zeta_t^j \leftarrow \varepsilon \cdot \zeta_t^j / |\zeta_t^j|_2$;
11:         $X_t^j \leftarrow X_{t-1}^j + \alpha_t V_{t-1}^j$;
12:         $W_{t-1}^j \leftarrow \delta \cdot Y_{t-1}^j + (1 - \delta) \cdot Y_{t-1}$;
13:         $V_t^j \leftarrow V_{t-1}^j + \alpha_t \cdot W_{t-1}^j + \beta_t \cdot X_{t-1}^j + \gamma_t \cdot X_t^j + \alpha_t \cdot \zeta_t^j$;
14:         /* $Y(t)$ update for the $j^{th}$ SHE$^2$ thread*/
15:         **if** $f(X_t^j) \leq f(Y_{t-1}^j)$ **then**
16:             $Y_t^j \leftarrow X_t^j$;
17:         **else**
18:             $Y_t^j \leftarrow Y_{t-1}^j$;
19:         **end if**
20:     **end for**
21:     /* Local Minimum Update*/
22:         $\underset{\forall X \in \{Y_t^1, Y_t^2, \ldots, Y_t^N\}}{\text{argmin}} \quad f(X) \rightarrow Y_t$;
23: **end for**
24: **return** $Y_T$;

---

### 4.3 APPROXIMATION ANALYSIS FOR P-SHE$^2$

As $\alpha_t \rightarrow 0$, we can rewrite the sequences $X_t^j$, $W_t^j$ and $V_t^j$ described in lines 9–12 of Algorithm 2 as the following Ordinary Differential Equations (ODEs) of $X^j(t)$, $W^j(t)$ and $V^j(t)$:

$$
\begin{cases}
\dfrac{X_t^j - X_{t-1}^j}{\alpha_t} = V_{t-1}^j & \overset{\alpha_t \rightarrow 0}{\Longrightarrow} \dot{X}(t) = V^j(t), \\[2mm]
W_{t-1}^j = \delta Y_{t-1}^j + (1 - \delta) Y_{t-1} & \overset{\alpha_t \rightarrow 0}{\Longrightarrow} W^j(t) = \delta Y^j(t) + (1 - \delta) Y(t), \\[2mm]
\dfrac{V_t^j - V_{t-1}^j}{\alpha_t} = -\dfrac{3(X_t^j - X_{t-1}^j)}{t \cdot \alpha_t} & \overset{\alpha_t \rightarrow 0}{\Longrightarrow} \dot{V}^j(t) = -\dfrac{3}{t} \dot{X}^j(t) \\[2mm]
\quad + (W_{t-1}^j - X_{t-1}^j) + \zeta_t^j & \quad - Q_\delta(X^j(t), Y^j(t), Y(t)) + \zeta^j(t),
\end{cases}
\tag{8}
$$

where $\zeta^j(t)$ refers to the continuous-time dynamics of sequence $\zeta_1^j, \zeta_2^j, \ldots, \zeta_T^j$ and $|\zeta^j(t)|_2 = \varepsilon$ for every time $t$. Through combining above three ODEs and Eq. 8, we can obtain the ODE of $X(t)$ as:

$$
\ddot{X}^j(t) + \frac{3}{t} \dot{X}^j(t) + Q_\delta(X^j(t), Y^j(t), Y(t)) - \zeta^j(t) = 0.
\tag{9}
$$

Using same the settings, we can conclude that $X_t^j$ would have similar behavior as $X^i(t)$ (for $1 \leq i \leq N$ in Eq 7). Thus, **Algorithm 2** can be viewed as a discrete-time approximation of dynamical systems in Eq 7. Since the sequence $Y_t$ always exploits the minimum point that has been found by all $N$ threads at every time $t$, we can use the algorithm output $Y_T$ as the minimizer of $f(x)$.

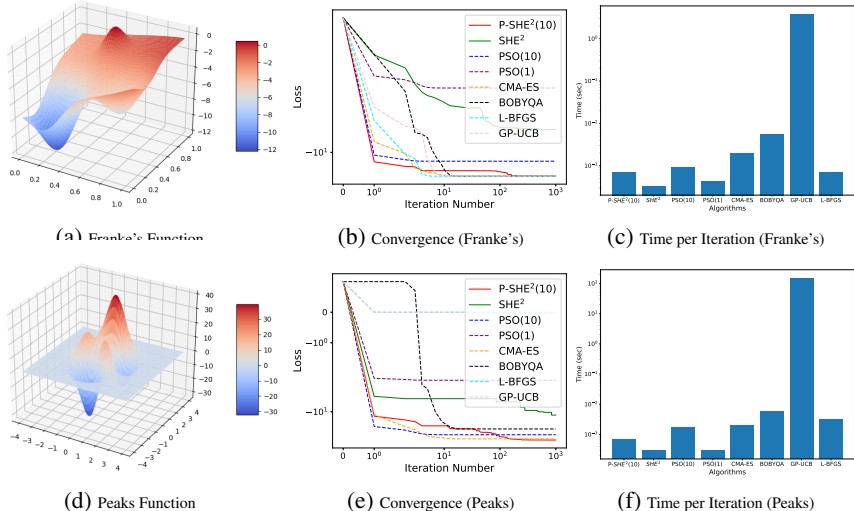

Figure 2: Performance Comparison using Nonconvex Functions

## 4.4 Connection to Existing Solutions

The proposed P-SHE$^2$ algorithm can be viewed as a particle swarm optimizer Kennedy (2011) with inverse-scale step-size settings. Compared to Particle Swarm, which usually adopts constant step-size settings (i.e., $\alpha_t$, $\beta_t$ and $\gamma_t$ are fixed as a constant value), P-SHE$^2$ proposes to use a small $\alpha_t$, while setting $\beta_t = -(\alpha_t + 3/t)$ and $\gamma_t = 3/t$ for each (the $t^{th}$) iteration. Such settings help the optimizer approximates to the Nesterov's scheme, so as to enjoy faster convergence speed, under certain assumption. In terms of contribution, our research made as yet an rigorous analysis for Particle Swarm through linking it to to Nesterov's scheme Nesterov (2013); Su et al. (2014).

## 5 Experiments and Empirical Validation

We provide three sets of experiments to validate our algorithms. In the first set of experiments, we demonstrate the performance of SHE$^2$ and P-SHE$^2$ to minimize two non-convex functions through the comparisons to a set of DFO optimizers, including Gaussian Process optimization algorithms (GP-UCB) (Martinez-Cantin, 2014), Powell's BOBYQA methods (Powell, 2009), Limited Memory-BFGS-B (L-BFGS) (Zhu et al., 1997), Covariance Matrix Adaptation Evolution Strategy (CMA-ES) (Hansen et al., 2003), and Particle Swarm optimizer (PSO) Kennedy (2011). For the second set of experiments, we use the same set of algorithms to train logistic regression (Lee et al., 2006) and support vector machine (Cortes & Vapnik, 1995) classifiers, on top of benchmark datasets, for supervised learning tasks. In the third set, we use P-SHE$^2$ to optimize the hyper-parameters of ResNet-50 for the performance tuning on Flower 102 and MIT Indoor 67 benchmark datasets under transfer learning settings.

### 5.1 Nonconvex Function Minimization

Figure 2 presents the performance comparison between P-SHE$^2$, SHE$^2$ and the baseline algorithms using two 2D benchmark nonconvex functions–Franke's function and Peaks function. Figure 2.a and c present the landscape of these two functions, while Figure 2.b and d present the performance evaluation of P-SHE$^2$, SHE$^2$ and baseline algorithms on these two functions. All these algorithms are tuned with best parameters and evaluated for 20 times, while averaged performance is presented. Specifically, we illustrate how these algorithms would converge with increasing number of iterations. Obviously, on Franke's function, only P-SHE$^2$(10), i.e., the P-SHE$^2$ algorithm with $N = 10$ search threads, CMA-ES, GP-UCB algorithms and BOBYQA converge to the global minimum, while the rest of algorithms, including SHE$^2$ and PSO, converge to the local minimum. Though P-SHE$^2$(10) needs more iterations to converge to the global minimum, its per iteration time consumption is significantly lower than the other three convergeable algorithms (shown in Figure 2.e). The same comparison result can be also observed from the comparison using Peaks function (in Figures 2.b and 2.c, only CMA-ES and P-SHE$^2$(10) converge to global minimum in the given number of iterations). Compared P-SHE$^2$(10) to PSO(10), they both use 10 search threads with the

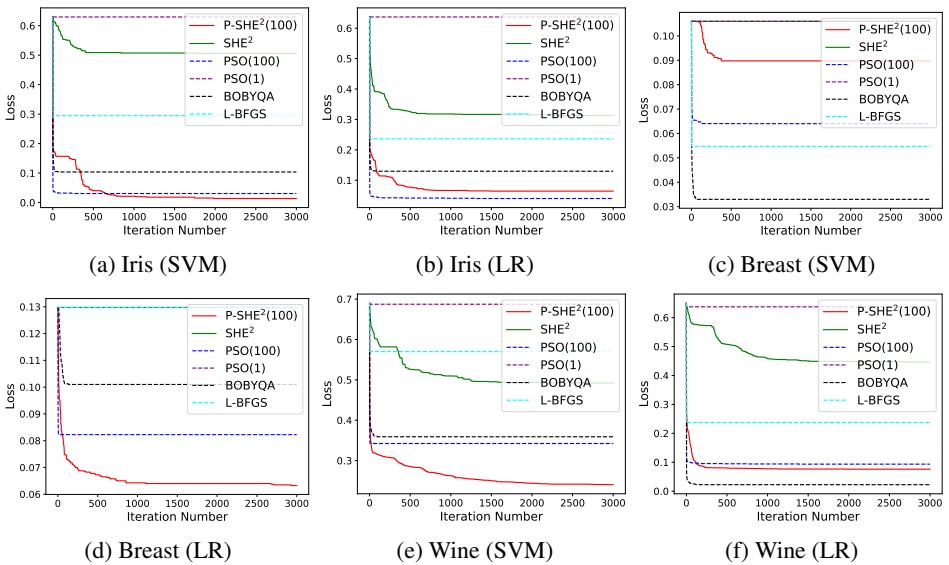

(a) Iris (SVM)      (b) Iris (LR)      (c) Breast (SVM)

(d) Breast (LR)      (e) Wine (SVM)      (f) Wine (LR)

Figure 3: Loss Minimization via DFO using Benchmark Datasets

same computational/memory complexity, while P-SHE$^2$(10) converges much faster than PSO(10). The same phenomena can be also observed from the comparison between SHE$^2$ and PSO(1), both of which search with single thread. We can suggest that the adaptive step-size settings inherent from Nesterove's scheme accelerate the convergence speed.

Table 1: Accuracy Comparison for Parameter Optimization on Benchmark Datasets

| Algorithms | Iris | | Wine | | Breast Cancer | |
|---|---|---|---|---|---|---|
| | LR | SVM | LR | SVM | LR | SVM |
| P-SHE$^2$(100) | **0.952 ± 0.023** | **0.987 ± 0.037** | **0.967 ± 0.034** | **0.961 ± 0.027** | **0.980 ± 0.014** | **0.982 ± 0.015** |
| Derivative-Free Solutions | | | | | | |
| SHE$^2$ | 0.493 ± 0.113 | 0.571 ± 0.065 | 0.628 ± 0.135 | 0.689 ± 0.138 | 0.848 ± 0.079 | 0.830 ± 0.112 |
| PSO(100) | 0.920 ± 0.059 | 0.933 ± 0.059 | 0.922 ± 0.047 | 0.900 ± 0.045 | 0.956 ± 0.028 | 0.947 ± 0.014 |
| PSO(1) | 0.361 ± 0.060 | 0.305 ± 0.034 | 0.355 ± 0.035 | 0.326 ± 0.015 | 0.608 ± 0.320 | 0.544 ± 0.211 |
| BOBYQA | 0.793 ± 0.118 | 0.933 ± 0.059 | 0.944 ± 0.042 | 0.953 ± 0.019 | 0.959 ± 0.022 | 0.869 ± 0.062 |
| L-BFGS | 0.747 ± 0.110 | 0.600 ± 0.202 | 0.772 ± 0.057 | 0.828 ± 0.093 | 0.815 ± 0.008 | 0.947 ± 0.014 |
| Derivative-based Solution | | | | | | |
| SGD* | 0.950 ± 0.026 | 0.930 ± 0.038 | 0.975 ± 0.011 | 0.981 ± 0.016 | 0.965 ± 0.008 | 0.897 ± 0.041 |

## 5.2 PARAMETER OPTIMIZATION FOR SUPERVISED LEARNING

We use above algorithms to train logistic regression (LR) and SVM classifiers using Iris (4 features, 3 classes and 150 instances), Breast (32 features, 2 classes and 569 instances) and Wine (13 features, 3 classes and 178 instances) datasets. We treat the loss functions of logistic regression and SVM as black-box functions and parameters (e.g., projection vector $\beta$ for logistic regression) as optimization outcomes. Note that the number of parameters for multi-class ($\#class \geq 3$) classification is $\#class \times \#features$, e.g., 39 for wine data. We don't include GP-UCB in the comparison, as it is extremely time-consuming to scale-up in high-dimensional settings.

Figure 3 demonstrates how loss function could be minimized by above algorithms with iterations by iterations. For both classifiers on all three datasets, P-SHE$^2$(100)–the P-SHE$^2$ algorithms with $N = 100$ search threads, outperforms all rest algorithms with the most significant loss reduction and the best convergence performance. We also test the accuracy of trained classifiers using the testing datasets. Table. 1 shows that both classifiers trained by P-SHE$^2$(100) enjoys the best accuracy among all above DFO algorithms and the accuracy is comparable to those trained using gradient-based optimizers. All above experiments are carried out under 10-folder cross-validation. Note that the accuracy of the classifiers trained by P-SHE$^2$ is closed to/or even better than some fine-tuned gradient-based solutions (Dogan & Tanrikulu, 2013; Roscher & Förstner, 2009).

## 5.3 HYPERPARAMETER OPTIMIZATION FOR DEEP NEURAL NETWORKS

To test the performance of P-SHE$^2$ for derivative-free optimization with noisy black-box function evaluation, We use P-SHE$^2$ to optimize the hyper-parameter of ResNet-50 networks for Flower

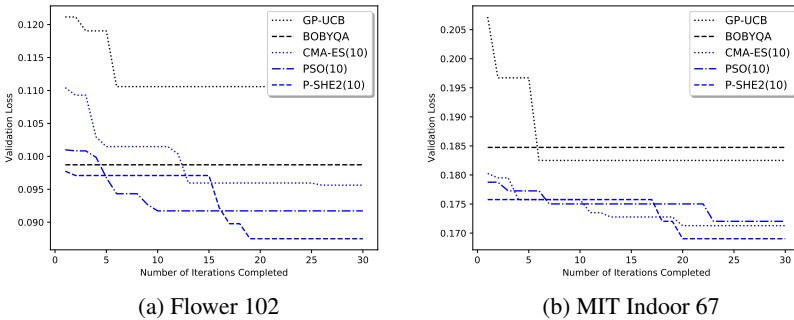

(a) Flower 102            (b) MIT Indoor 67

Figure 4: Hyper-Parameter Optimization for ResNet-50 on Benchmark Datasets

102 recognition (Nilsback & Zisserman, 2008) and MIT Indoor 67 classification (Quattoni & Torralba, 2009) tasks. The two networks are pre-trained using ImageNet (Krizhevsky et al., 2012) and Place365 datasets (Zhou et al., 2014), respectively. Specifically, we design a black-box function to package the training procedure of the ResNet-50, where 12 continuous parameters, including the learning rate of procedure, type of optimizers (after simple discretization), the probabilistic distribution of image pre-processing operations for randomized data augmentation and so on, are interfaced as the input of the function while the validation loss of the network is returned as the output. We aim at searching the optimal parameters with the lowest validation loss. The experiments are all based on a Xeon E5 cluster with many available TitanX, M40x8, and 1080Ti GPUs.

Our experiments compare P-SHE$^2$ with a wide range of solvers and hyper-parameter tuning tools, including PSO, CMA-ES (Loshchilov & Hutter, 2016), GP-UCB (Joy et al., 2016) and BOBYQA under the same pre-training/computing settings. Specifically, we adopt the vanilla implementation of GP-UCB and BOBYQA (with single search thread), while P-SHE$^2$, PSO and CMA-ES are all with 10 search threads for parallel optimization. The experimental results show that all these algorithms can well optimize the hyer-parameters of ResNet for the better performance under the same settings, while P-SHE$^2$ has ever searched the hyperparameters with the lowest validation loss in our experiments (shown in Figure 4). Due to the restriction of PyBOBYQA API, we can only provide the function evaluation of the final solution obtained by BOBYQA as a flatline in Figure 4. In fact, P-SHE$^2$, PSO and CMA-ES may spend more GPU hours than GP-UCB and BOBYQA due to the parallel search. For the fair comparison, we also evaluate GP-UCB and BOBYQA with more than 100 iterations til the convergence, where GP-UCB can achieve 0.099854 validation error (which is comparable to the three parallel solvers) for Flower 102 task. Note that we only claim that P-SHE$^2$ can be used for hyerparameter optimization with decent performance. We don't intend to state that P-SHE$^2$ is the best for hyerparameter tuning, as the performance of the three parallel solvers are sometimes randon and indeed close to each other.

## 6 DISCUSSION AND CONCLUSION

In this paper, we present SHE$^2$ and P-SHE$^2$ – two derivative-free optimization algorithms that leverage a Hamiltonian exploration and exploitation dynamical systems for black-box function optimization. Under mild condition SHE$^2$ algorithm behaves as a discrete-time approximation to a Nestereov's scheme ODE (Su et al., 2014) over the quadratic trust region of the blackbox function. Moreover, we propose P-SHE$^2$ to further accelerate the minimum search through parallelizing multiple SHE$^2$-alike search threads with simple synchronization.

Compared to the existing trust region methods, P-SHE$^2$ uses multiple quadratic trust regions with multiple (coupled) stochastic Hamiltonian dynamics to accelerate the exploration-exploitation processes, while avoiding the needs of Hessian matrix estimation for quadratic function approximation. Instead of interpolating sampled points in one quadratic function, P-SHE$^2$ *defacto* constructs one quadratic surrogate (with identity Hessian) for each sampled point and leverages parallel search threads with parallel black-box function evaluation to boost the performance. Experiment results show that P-SHE$^2$ can compete a wide range of DFO algorithms to minimize nonconvex benchmark functions, train supervised learning models via parameter optimization, and fine-tune deep neural networks via hyperparameter optimization.

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

## A  PROOF OF THE CONVERGENCE OF SHE$^2$ DYNAMICAL SYSTEM

Our goal is to show that in the system (4) we have $X(t) \to x^*$ as $t \to \infty$, where $x^*$ is a local minimum point of the landscape function $f(x)$.

**Definition 4.** *We say the point $x^*$ is a local minimum point of the function $f(x)$ if and only if $f(x^*) \leq f(x)$ for any $x \in U(x^*)$, where $U(x^*)$ which is any open neighborhood around the point $x^*$.*

Let us first remove the noise $\zeta(t)$ in our system (4). Thus we obtain the following deterministic dynamical system $(X^0(t), Y^0(t))$:

$$\begin{cases} (a) & \ddot{X}^0(t) + \dfrac{3}{t}\dot{X}^0(t) + \dfrac{\partial}{\partial X}Q(X^0(t), Y^0(t)) = 0 \,, \\ (b) & Y^0(t) = \arg\min_{x \in \mathbb{T}(t)} f(x) \,, \ \mathbb{T}(t) = \{X^0(\tau) : \tau \in (0, t]\} \,. \end{cases} \tag{10}$$

In the equations (a) and (b) of (10), the pair of processes $(X^0(t), Y^0(t))$ is a pair of coupled processes. In the next Lemma, we show that $X^0(t)$ converges to the minimum point of $f(x)$ *along the trajectory of* $X^0(t)$ as $t \to \infty$.

**Lemma 1.** *For the deterministic system* (10)*, we have that $\|X^0(t) - Y^0(t)\|_2 \to 0$ as $t \to \infty$, so that the coupled process $(X^0(t), Y^0(t))$ converges to diagonal.*

*Proof.* Set $\dot{X}^0(t) = P(t)$, we can write equation (a) in (10) in Hamiltonian form

$$\begin{cases} \dot{X}^0(t) = P(t) \,, \\ \dot{P}(t) = -\dfrac{3}{t}P(t) - \dfrac{\partial}{\partial X}Q(X^0(t), Y^0(t)) \,. \end{cases} \tag{11}$$

Set $H(X, Y, P) = \dfrac{P^2}{2} + Q(X, Y)$. Then we have

$$\begin{aligned} & \frac{d}{dt}H(X^0(t), Y^0(t), P(t)) \\ =\ & P(t) \cdot \dot{P}(t) + \frac{\partial}{\partial X}Q(X^0(t), Y^0(t)) \cdot \dot{X}^0(t) + \frac{\partial}{\partial Y}Q(X^0(t), Y^0(t)) \cdot \dot{Y}^0(t) \\ =\ & P(t) \cdot \left(-\frac{3}{t}P(t) - \frac{\partial}{\partial X}Q(X^0(t), Y^0(t))\right) + \frac{\partial}{\partial X}Q(X^0(t), Y^0(t)) \cdot P(t) \\ & + \frac{\partial}{\partial Y}Q(X^0(t), Y^0(t)) \cdot \dot{Y}^0(t) \\ =\ & -\frac{3}{t}\|P(t)\|_2^2 + \frac{\partial}{\partial Y}Q(X^0(t), Y^0(t)) \cdot \dot{Y}^0(t) \,. \end{aligned} \tag{12}$$

As we have $Q(X, Y) = \dfrac{1}{2}\|X - Y\|_2^2$, we see from (12) that we have

$$\frac{d}{dt}H(X^0(t), Y^0(t), P(t)) = -\frac{3}{t}\|P(t)\|_2^2 - (X^0(t) - Y^0(t)) \cdot \dot{Y}^0(t) \,. \tag{13}$$

Notice that by our construction part (b) of the coupled process (10), we have $f(Y^0(t)) \leq f(X^0(s))$ for $0 \leq s \leq t$. If $X^0(t) - Y^0(t) = 0$, then $(X^0(t) - Y^0(t)) \cdot \dot{Y}^0(t) = 0$. If $X^0(t) - Y^0(t) \neq 0$, then we see that $Y^0(t) = X^0(s_0)$ for some $0 \leq s_0 < t$, and $f(X^0(s_0)) < f(X^0(t))$. By continuity of the trajectory of $X^0(t)$ as well as the function $f(x)$, we see that in this case $\dot{Y}^0(t) = 0$, so that $(X^0(t) - Y^0(t)) \cdot \dot{Y}^0(t) = 0$. Thus we see that (13) actually gives

$$\frac{d}{dt}H(X^0(t), Y^0(t), P(t)) = -\frac{3}{t}\|P(t)\|_2^2 \leq 0 \,. \tag{14}$$

From here, we know that $H(X^0(t), Y^0(t), P(t))$ keeps decaying until $P(t) \to 0$ and $\|X^0(t) - Y^0(t)\|_2 \to 0$, as desired. $\qquad\square$

Since $f(Y^0(t)) \leq \min_{0 \leq s \leq t} f(X^0(s))$, Lemma 1 tells us that as $t \to \infty$, the deterministic process $X^0(t)$ in (10) approaches the minimum of $f$ along the trajectory traversed by itself. Let us now add the noise $\zeta(t)$ to part (a) of (10), so that we come back to our original system (4). We would like to argue that with the noise $\zeta(t)$, we actually have $\lim_{t \to \infty} \min_{0 \leq s \leq t} f(X(s)) = f(x^*)$, and thus $X(t) \to x^*$ when $t \to \infty$ as desired.

**Lemma 2.** *For the process $X(t)$ in part (a) of the system (4), we have $X(t) \to x^*$ as $t \to \infty$, where $x^*$ is a local minimum of the landscape function $f(x)$.*

*Proof.* We first notice that we have $\zeta(t) = \varepsilon \frac{\xi_t}{\|\xi_t\|_2}$, where $\xi_t \sim \mathcal{N}(0, I)$ is a sequence of i.i.d. normal, so that $\|\zeta(t)\|_2 = \varepsilon$, $\mathbb{E}\zeta(t) = 0$. Viewing (10) as a small random perturbation (see (Freidlin & Wentzell, 2012, Chapter 2, Section 1)) of the system (4) we know that for any $\delta > 0$ fixed, we have

$$\mathbb{P}\left( \max_{0 \leq t \leq T} \|X(t) - X^0(t)\|_2 \geq \delta \right) \to 0 \tag{15}$$

as $\varepsilon \to 0$. From here we know that the process $X(t)$ behaves close to $X^0(t)$ with high probability, so that by Lemma 1 we know that with high probability we have

$$\lim_{t \to \infty} \left( f(X(t)) - \min_{0 \leq s \leq t} f(X(s)) \right) = 0 . \tag{16}$$

Our next step is to improve the above asymptotic to $X(t) \to x^*$ as $t \to \infty$. Comparing with (16), we see that it suffices to show

$$\lim_{t \to \infty} \min_{0 \leq s \leq t} f(X(s)) = f(x^*) . \tag{17}$$

To demonstrate (17), we note that when $t$ is large, we can ignore in (4) the damping term $\frac{3}{t}\dot{X}(t)$ and obtain a friction-less dynamics

$$\begin{cases} (a): & \ddot{X}(t) + \frac{\partial}{\partial X}Q(X(t), Y(t)) + \zeta(t) = 0 , \\ (b): & Y(t) = X(\tau) \text{ and } \tau = \underset{\tau \in (0,t]}{\operatorname{argmin}} f(X(\tau)) . \end{cases} \tag{18}$$

Combining Lemma 1, (15) and $\frac{\partial}{\partial X}Q(X,Y) = X - Y$ we further see that the term $\frac{\partial}{\partial X}Q(X,Y)$ also contribute little in (18). Thus part (a) of (18) reduces to a very simple equation

$$\ddot{X}(t) + \zeta(t) = 0 . \tag{19}$$

Equation (19) enables the process $X(t)$ to explore locally in an ergodic way its neighborhood points, so that if (17) is not valid, then $X(t + dt)$ will explore a nearby point at which $f(X(t + dt))$ is less that $\min_{0 \leq s \leq t} f(X(t))$, and thus will move to that point. This leads to a further decay in the value of $f(X(t))$, which demonstrates that in he limit $t \to \infty$ we must have (17), and the Lemma concludes. $\qquad\square$

Summarizing, we have the Theorem 1.

## B  DISCUSSIONS ABOUT THE RATE OF CONVERGENCE OF SHE$^2$ DYNAMICAL SYSTEM

Here we provide a short discussion on the convergence rate of the algorithm SHE2. In the previous appendix we have demonstrated that the system (4) converges via two steps. Step 1 in Lemma 1 shows that the differential equation modeling Nesterov's accelerated gradient descent (see Su et al. (2014)) helps the process $X(t)$ to "catch up" with the minimum point $Y(t)$ on its path. Step 2 in Lemma 2 shows that when $t \to \infty$ the noise term $\zeta(t)$ helps the process $X(t)$ to reach local

| Name | Tpye & Range |
|---|---|
| Hyperparameters related to Training | |
| Batch Size | Integer, [16,32] |
| Learning Rate | Float, [0.0005, 0.002] |
| Decay Rate of Learning Rate | Float, [0.7, 0.9] |
| Regularization coefficient | Float, $[1 \times e^{-4}, 2 \times e^{-3}]$ |
| Hyperparameters related to Data Augmentation | |
| Resize Type | Boolean, {'keep aspect ratio', 'ignore aspect ratio'} |
| Crop Type | Discrete, {'none', 'random', 'center'} |
| Horizontal Flip | Float (probability), [0,0.5] |
| Vertical Flip | Float (probability), [0,0.5] |
| Rotate | Float, [0,20] |
| Color Jitter | Float, [0,0.005] |
| Cutout | Boolean, {'cutout', 'not cutout'} |
| Tencrop | Boolean, {'tencrop', 'not tencrop'} |

Table 2: Definitions of Hyperparameters

minimum point of $f(x)$. According to the general theory regrading Nesterov's accelerated gradient method (see Nesterov and Su et al. (2014)), we naturally expect that the first step has convergence rate

$$f(X(t)) - f(Y(t)) \leq \mathcal{O}(1/t^2) \ .$$

For the second step, we see that at each iteration the process is searching for a direction where the value of the function $f(X(t))$ will be decreasing, using the random inputs $\zeta(t)$. This leads to an additional time cost of $\mathcal{O}(1/\varepsilon)$ to reach the local minimum point $x^*$.

**Remark 3.** *In the Nesterov's algorithm we actually ususally do not use* $\dfrac{3}{t}$*, but the following scheme:*

$$x_k = y_{k-1} - s\nabla f(y_{k-1}) \ , \ y_k = x_k + \frac{k-1}{k+2}(x_k - x_{k-1}) \ ,$$

*where $s > 0$ is the small stepsize. Since* $\dfrac{k-1}{k+2} = \dfrac{(k+2)-3}{k+2} \approx 1 - \dfrac{3}{k}$ *as $k$ is large, we obtain our* $\dfrac{3}{t}$*. So in real Nesterov's method one can even choose the* $\dfrac{3}{t}$ *to be* $\dfrac{3}{t+2}$*.*

## C  HYERPARAMETERS FOR RESNET-50 TRAINING

The hyprparameters for ResNet-50 training on both MIT Indoor 67 and flower 102 datasets are listed in Tabl 2. The range of all hyperparameters are uniformly mapped to the range of $[-10, 10]$ for optimization. For example, the Boolean hyperparameter "Resize type" is mapped to $[-10, 0) \rightarrow$ "keep aspect ratio" and $[0, 10) \rightarrow$ "ignore aspect ratio", respectively.

