# OpenReview forum: "SHE2: Stochastic Hamiltonian Exploration and Exploitation for Derivative-Free Optimization"
_ICLR.cc/2019/Conference_

### Official Review · AnonReviewer1 · 2018-10-28
**Interesting idea but the analysis and the writing need to be improved**

**Rating:** 3
**Confidence:** 5

**Review:**

This paper suggests a continuous-time framework consisting of two coupled processes in order to perform derivative-free optimization. The first process optimizes a surrogate function, while the second process updates the surrogate function. This continuous-time process is then discretized in order to be run on various machine learning datasets. Overall, I think this is an interesting idea as competing methods do have high computational complexity costs. However, I’m not satisfied with the current state of the paper that does not properly discuss notions of complexity of their own method compared to existing methods.

1) “The computational and storage complexity for (convex) surrogates is extremely high.” The discussion in this paragraph is too superficial and not precise enough.
a) First of all, the authors only discuss quadratic models but one can of course use linear models as well, see two references below (including work by Powell referenced there):
Chapter 9 in Nocedal, J., & Wright, S. J. (2006). Numerical optimization 2nd.
Conn, A. R., Scheinberg, K., & Vicente, L. N. (2009). Global convergence of general derivative-free trust-region algorithms to first-and second-order critical points. SIAM Journal on Optimization, 20(1), 387-415.
I think this discussion should also be more precise, the authors claim the cost is extremely high but I would really expect a discussion comparing the complexity of this method with the complexity of their own approach. As discussed in Nocedal (reference above) the cost of each iteration with a linear model is O(n^3) instead of O(n^4) where n is the number of interpolation points. Perhaps this can also be improved with more recent developments, the authors should do a more thorough literature review.
b) What is the complexity of the methods cited in the paper that rely on Gaussian processes?
(including (Wu et al., 2017) and mini-batch (Lyu et al., 2018)).


2) “The convergence of trust region methods cannot be guaranteed for high-dimensional nonconvex DFO”
Two remarks: a) This statement is incorrect as there are global convergence guarantees for derivative-free trust-region algorithms, see e.g.
Conn, A. R., Scheinberg, K., & Vicente, L. N. (2009). Global convergence of general derivative-free trust-region algorithms to first-and second-order critical points. SIAM Journal on Optimization, 20(1), 387-415.
In chapter 10, you will find global convergence guarantees for both first-order and second-order critical points.
b) The authors seem to emphasize high-dimensional problems although the convergence guarantees above still apply. For high-order models, the dimension does have an effect, please elaborate on what specific comment you would like to make. Finally, can you comment on whether the lower bounds derived by Jamieson mentioned depend on the dimension.

3) Quadratic loss function
The method developed by the authors rely on the use of a quadratic loss function. Can you comment on generalizing the results derived in the paper to more general loss functions? It seems that the computational complexity wouldn’t increase as much as existing DFO methods. Again, I think it would be interesting to give a more in-depth discussion of the complexity of your approach.

4) Convergence rate
The authors used a perturbed variant of the second-order ODE defined in Su et al. 2014. The noise added to the ODE implies that the analysis derived in Su et al. 2014 does not apply as is. In order to deal with the noise the authors show that unbiased noise does not affect the asymptotic convergence. I think the authors could get strong non-asymptotic convergence results. In a nutshell, one could use tools from Ito calculus in order to bound the effect of the noise in the derivative of the Hamiltonian used in Lemma 1. See following references:
Li, Q., Tai, C., et al. (2015). Stochastic modified equations and adaptive stochastic gradient algorithms. arXiv preprint arXiv:1511.06251.
Krichene, W., Bayen, A., and Bartlett, P. L. (2015). Accelerated mirror descent in continuous
and discrete time. In Advances in neural information processing systems, pages 2845–2853.
Of course, the above works rely on the use of derivatives but as mentioned earlier, one should be able to rely on existing DFO results to prove convergence. If you check Chapter 2 in the book of Conn et al. (see reference above), you will see that linear interpolation schemes already offer some simple bounds on the distance between the true gradient of the gradient of the model (assuming Lipschitz continuity and differentiability).

5) Noise
“The noise would help the system escape from an unstable stationary point in even shorter time”
Please add a relevant citation. For isotropic noise, see
Ge, R., Huang, F., Jin, C., and Yuan, Y. Escaping from saddle points-online stochastic gradient for tensor decomposition.
Jin, C., Netrapalli, P., and Jordan, M. I. Accelerated gradient descent escapes saddle points faster than gradient descent. arXiv preprint arXiv:1711.10456,

6) Figure 2
Instead of having 2 separate plots for iteration numbers and time per iteration, why don’t you combine them to show the loss vs time. This would make it easier for the reader to see the combined effect.

7) Empirical evaluation
a) There are not enough details provided to be able to reproduce the experiments. Reporting the range of the hyperparameters (Table 2 in the appendix) is not enough. How did you select the hyperparameters for each method? Especially step-size and batch-size which are critical for the performance of most algorithms.
b) I have to admit that I am not extremely familiar with common experimental evaluations used for derivative-free methods but the datasets used in the paper seem to be rather small. Can you please justify the choice of these datasets, perhaps citing other recent papers that use similar datasets?

8) Connection to existing solutions
The text is quite unclear but the authors seem to claim they establish a rigorous connection between their approach and particle swarm (“In terms of contribution, our research made as yet an rigorous analysis for Particle Swarm”). This however is not **rigorously** established and needs further explanation. The reference cited in the text (Kennedy 2011) does not appear to make any connection between particle swarm and accelerated gradient descent. Please elaborate.

9) SGD results
Why are the results for SGD only reported in Table 1 and not in the figure? Some results for SGD are better than for P-SHE2 so why are you bolding the numbers for P-SHE2?
It also seem surprising that SGD would achieve better results than the accelerated SGD method. What are the possible explanations?

10) Minor comments
- Corollaries 1 and 2 should probably be named as theorems. They are not derived from any other theorem in the paper. They are also not Corollaries in Su et al. 2014.
- Corollary 2 uses both X and Z.
- Equation 5, the last equation with \dot{V}(t): there is a dot missing on top of the first X(t)
“SHE2 should enjoy the same convergence rate Ω(1/T) without addressing any further assumptions” => What do you mean by “should”?
- There are **many** typos in the text!! e.g. “the the”, “is to used”, “convergeable”,... please have someone else proofread your submission.

---

### Official Review · AnonReviewer3 · 2018-11-02
**Concerns about clarity, and I'm confused about the inclusion of NGD and L-BFGS as "derivative free"**

**Rating:** 3
**Confidence:** 3

**Review:**

Overall, I could potentially be persuaded to accept this paper given a relatively favorable comparison to some other blackbox optimization algorithms, but I have some serious issues about clarity and some technical details that seem wrong to me (e.g., the inclusion of L-BFGS as a "derivative free" baseline, and the authors' method outperforming derivative based methods at optimizing convex loss functions).

I'd like to start by focusing on a few of the results in section 5.2 specifically.
In this section, you compare your method and several baselines on the task of training logistic regression and
SVM models. Given that these models have convex loss functions, it is almost inconceivable to me that methods like L-BFGS and SGD (at least with decent learning rates) would perform worse than gradient free optimization algorithms,
as both L-BFGS and SGD should clearly globally optimize a convex loss. I am also generally confused by the inclusion
of L-BFGS as an example of a derivative free optimization problem. Are you using L-BFGS with search directions
other than the gradient or something as a baseline? I think the exact setup here may require substantially more explanation.

The primary other issue I'd like to discuss is clarity. While I think the authors do a very good job
giving formal definitions of their proposed methods, the paper would massively benefit from some additional
time spent motivating the authors' approach. As a primary example, definition 2 is extremely confusing. I felt it wasn't as well motivated as it could have been given that it is  the central contribution of the paper. You reference an "exploration process" and an "exploitation process" that "were shown in Eq. 4," but equation four is the next equation that directly jumps in to using these two processes X(t) and Y(t). These two processes are very vaguely
defined in the definition. For example, I understand from that definition that Y(t) tracks the current min value, but even after reading the remainder of the paper I am still not entirely sure I understand the purpose of X(t). Perhaps
the paper assumes a detailed understanding on the readers' part of the work in Su et al. (2014), which is cited
repeatedly throughout the method section?

To be concrete, my recommendation to the authors would be to substantially shorten the discussion in the paper
before section 3, provide background information on Su et al., 2014 if necessary, and spend a substantially
larger portion of the paper explaining the derivation of SHE2 rather than directly presenting it as an ODE
that immediately introduces its own notation. In the algorithm block, the underlying blackbox function
is only evaluated in the if statement on line 9 -- can the authors explain intuitively how their surrogate
model evolves as a result of the Y_{t} update?

In addition to these concerns, some of the claims made in the method section seem strange or even wrong to me,
and I would definitely like to see these addressed in some way. Here is a list of a few concerns I have
along this line:

- A few of the citations you've given as examples of derivative free optimization are confusing.
You cite natural gradient methods and L-BFGS as two examples, but natural gradient descent involves preconditioning
the gradient with the inverse Fisher information matrix, and is therefore typically not derivative
free. You give Gaussian process surrogate models as an example of a convex surrogate, but GPs
in general do not lead to convex surrogates save for with very specific kernels that are not
often used for Bayesian optimization.

- In the background, it reads to me like you define GP based Bayesian optimization as a quadratic
based trust region method. This seems strange to me. Trust region methods do involve quadratic surrogates,
but my understanding is that they are usually local optimization schemes where successive local quadratic
approximations are made for each step. GP based Bayesian optimization, by contrast, maintains a global
surrogate of the full loss surface, and seeks to perform global optimization.

- Equation 3 defines the squared norm \frac{1}{2}||X-Y||^{2}_{2} as the "Euclid[ean] distance".
Based on the following derivatives, I assume this is intended to be kept as
the squared Euclidean distance (with the 1/2 term included for derivative simplicity).

---

### Official Review · AnonReviewer2 · 2018-11-04
**Additional benchmarking is needed**

**Rating:** 4
**Confidence:** 4

**Review:**

Derivative-free optimization is not a novel domain and your work could benefit from some accepted benchmarking practices. For instance, you can consider the Black-Box Optimization Benchmarking (BBOB) Workshop and its COCO platform which was used to test many optimization algorithms including the ones mentioned in the paper.
Benchmarking on BBOB problems would probably reveal that your algorithm fails on non-separable ill-conditioned problems and even on problems like Rosenbrock (e.g., compared to BOBYQA). The results for other algorithms can be downloaded, you don't need to rerun them. BBOB's computational budget can be as low as 1 function evaluation.

Extended review of Update 17 Nov:
I would like to note that I liked the fact that you used several optimization algorithms in your comparison. To my best understanding, several algorithms shown in Figure 3 (e.g., BOBYQA and L-BFGS) would benefit from restarts and it is fairly common to use restarts when the computational budget allows it (it seems to be the case for Figure 3).

The results shown in Figure 4 are hard to trust because it does not seem that we observe mean/median results but probably a single run where the results after 1 iteration are drastically different for different algorithms. For instance, after one iteration BOBYQA only tests its second DOE point. Here, again, the issue is that one iteration for BOBYQA is 1 function evaluation while it is several (10) function evaluations for other algorithms. In that scenario, it would be more fair to run BOBYQA with 10 different initializations as well.
I don't understand "Due to the restriction of PyBOBYQA API, we can only provide the function evaluation of the final solution obtained by BOBYQA as a flatline in Figure 4". At the point when your objective function (which is not part of PyBOBYQA API) is called, I suppose you can log everything you need.

---

### Meta-Review · Area_Chair1 · 2018-12-12
**Interesting contribution that is not quite ready for publication**

**Confidence:** 5
**Recommendation:** Reject

**Metareview:**

In general the reviewers found the work to be interesting and the results to be promising.  However, all the reviewers shared significant concerns about the clarity of the paper and the correctness of technical claims made.  This paper would significantly benefit from rewriting and restructuring the paper to improve clarity, better motivate the approach and provide more careful exposition of related work and technical claims.